# Complete Mitochondrial Genomes and Phylogenetic Analysis of Genus *Henricia* (Asteroidea: Spinulosida: Echinasteridae)

**DOI:** 10.3390/ijms25115575

**Published:** 2024-05-21

**Authors:** Maria Alboasud, Hoon Jeong, Taekjun Lee

**Affiliations:** 1Marine Biological Resource Institute, Sahmyook University, Seoul 01795, Republic of Korea; mariashihab1@gmail.com; 2Department of Convergence Science, Sahmyook University, Seoul 01795, Republic of Korea; 3Department of Animal Resources Science, Sahmyook University, Seoul 01795, Republic of Korea

**Keywords:** *Henricia*, sea star, mitogenome, phylogeny, taxonomy

## Abstract

The genus *Henricia* is known to have intraspecific morphological variations, making species identification difficult. Therefore, molecular phylogeny analysis based on genetic characteristics is valuable for species identification. We present complete mitochondrial genomic sequences of *Henricia longispina aleutica*, *H. reniossa*, and *H. sanguinolenta* for the first time in this study. This study will make a significant contribution to our understanding of *Henricia* species and its relationships within the class Asteroidea. Lengths of mitochondrial genomes of the three species are 16,217, 16,223, and 16,194 bp, respectively, with a circular form. These genomes contained 13 protein-coding genes, two ribosomal RNA genes, 22 transfer RNA genes, and a D-loop. The gene order and direction aligned with other asteroid species. Phylogenetic relationship analysis showed that our *Henricia* species were in a monophyletic clade with other *Henricia* species and in a large clade with species (*Echinaster brasiliensis*) from the same family. These findings provide valuable insight into understanding the phylogenetic relationships of species in the genus *Henricia*.

## 1. Introduction

The mitochondrial genome is a double-stranded circular molecule typically ranging from 14,000 bp to 17,000 bp in length. It contains a gene complement encoding 13 protein-coding genes (PCGs), 22 transfer RNA genes (tRNA), and two ribosomal RNA genes (rRNA) [1]. Mitogenome sequencing is widely employed to elucidate phylogenetic and evolutionary relationships among species [2,3]. The mitogenome provides valuable insight into rearrangement patterns and phylogenetic relationships [4]. The class Asteroidea de Blainville, 1830 (sea stars), belonging to the phylum Echinodermata Klein, 1778, represents the second most diverse class in this phylum, encompassing 1800 species categorized into 38 families [5]. These intriguing creatures inhabit oceans worldwide, with tropical Atlantic and Indo-Pacific regions showcasing the highest diversity across various depths [6]. Asteroids have garnered significant attention due to their unique chemical structures, particularly their steroid metabolites, which offer rich opportunities for exploration. Starfish are also renowned for their wide range of biological effects, including antiviral, cytotoxic, antifungal, antibacterial, anti-inflammatory, anticancer, analgesic, and neurogenic actions [7,8].

Genus *Henricia* Gray, 1840, is a large genus that belongs to the family Echinasteridae Verrill, 1870. *Henricia* is known for its wide-ranging geographical distribution and intraspecific morphological variations. These variations are due to differences in environmental conditions that can result in phenotypic plasticity, making it difficult to identify species within the genus [9,10,11]. *Henricia* species are distributed in cold water. They exhibit great diversity in high latitudinal regions and deep-sea settings in the North Pacific Ocean, the Atlantic Ocean, and the East Sea of Korea [12]. According to recent studies by Mah, *Henricia* is currently composed of 97 species [13], including 21 species reported in South Korea [14].

Many species within the genus *Henrica* share similar characteristics, making it difficult to identify them based on their appearance alone [9,12]. An alternative approach is to use molecular data such as DNA barcoding to simplify species diagnosis. This method has become more common when species diagnosis is challenging due to limited or variable diagnostic morphological characteristics [15,16,17,18]. In this study, we used a mitogenomic approach that could provide more accurate data than single gene analysis. To construct our phylogenetic analysis of *Henricia* species, we utilized complete mitochondrial genomes of three *Henricia* species and 39 other species within the class Asteroidea. By examining other species within the class Asteroidea, we aimed to gain a comprehensive understanding of inter-specific phylogenetic relationships.

## 2. Results

### 2.1. Mitogenome Features and AT/GC-Skew

We investigated the mitogenomes of three *Henricia* species, *H. longispina aleutica*, *H. reniossa*, and *H. sanguinolenta*. Our findings showed that the lengths of their mitogenomes were 16,217, 16,194, and 16,223 bp, respectively. Each of these mitogenomes included 13 PCGs, 22 tRNAs, and two rRNAs. The base composition analysis of *H. longispina aleutica* revealed the following percentages: A (36.2%), T (26.2%), C (24.1%), and G (13.5%). The complete mitochondrial genome of *H. longispina aleutica* exhibited a preference for GC content (37.5%). Additionally, all 13 PCGs, as well as tRNA and rRNA genes, demonstrate anti-G bias. Similarly, base composition analysis of *H. reniossa* showed the following percentages: A (35.4%), T (25.7%), C (25.0%), and G (13.9%). The complete mitochondrial genome of *H. reniossa* displays a preference for GC content (38.8%). All 13 PCGs, as well as the tRNA and rRNA genes, also demonstrated an anti-G bias. Lastly, the base composition analysis of *H. sanguinolenta* showed the following percentages: A (35.6%), T (25.8%), C (24.6%), and G (13.9%). The complete mitochondrial genome of *H. sanguinolenta* also exhibited a preference for GC content (38.5%). All 13 PCGs, tRNA, and rRNA genes also demonstrate an anti-G bias (Table 1).

### 2.2. PCG Characteristics

Mitochondrial genomes of the three *Henricia* species were analyzed. PCGs in *H. longispina aleutica* and *H. sanguinolenta* spanned 11,473 bp, while *H. reniossa* spanned 11,476 bp. NADH5 was the longest at 1878 bp, while ATP8 was the shortest at 165 bp in *H. longispina aleutica* and *H. sanguinolenta*. NADH5 was the longest PCG, consisting of 1881 bp, while ATP8 was the shortest at 165 bp in *H. reniossa* (Figure 1, Figure 2 and Figure 3, Table 2). These PCGs collectively encoded 3824 amino acids for *H. longispina aleutica* and *H. sanguinolenta* and 3825 amino acids for *H. reniossa*.

The majority of the PCG genes of *H. longispina aleutica*, *H. reniossa*, and *H. sanguinolenta* began with the start codon ATG (Table 2). However, NAD4L and NADH3 of *H. longispina aleutica* and *H. sanguinolenta* started with ATC and ATT, respectively. In *H. reniossa*, start codons for NAD4L, NADH3, NADH2, and NADH1 were ATC, ATT, GTG, and GTG, respectively. The most common stop codon in the three *Henricia* species was TAA. However, COX1 and NADH4 utilized TGA, and CYTB employed TAG in *H. longispina aleutica* and *H. sanguinolenta*. COX1 and CYTB utilized TGA as a stop codon in *H. reniossa* (Table 2). The positive strand of *H. longispina aleutica*, *H. reniossa*, and *H. sanguinolenta* mitogenome contained 10 PCGs, while the negative strand contained NADH1, NADH2, and NADH6.

### 2.3. Relative Synonymous Codon Usage

Codons with relative synonymous codon usage (RSCU) values above 1.00 indicated a positive codon usage bias (CUB). They were considered abundant codons. Those with RSCU values below 1.00 indicated a negative CUB. They were considered less abundant codons [19]. Among the 13 PCGs of *H. longispina aleutica*, *H. reniossa*, and *H. sanguinolenta*, the most frequently used codons encoded serine (3.2%) for *H. longispina aleutica*, serine (2.56%) for *H. reniossa*, and serine (3.2%) for *H. sanguinolenta* had the most frequently encoded amino acids. Alanine, arginine, glutamate, glycine, and glutamine were the lowest frequencies among the three *Henricia* species (Table 3).

### 2.4. Characteristics of rRNA and tRNA Genes

The analysis of *H. longispina aleutica*, *H. reniossa*, and *H. sanguinolenta* revealed that their mitochondrial genomes encoded two rRNA genes: 12S and 16S rRNA. The 12S rRNA gene was located on the positive strand, while the 16S rRNA gene was situated on the negative strand of all three *Henricia* species (Figure 1, Figure 2 and Figure 3, Table 2). The 12S rRNA gene had a length of 931, 922, or 928 bp. In contrast, the 16S rRNA gene spanned 1585, 1594, and 1607 bp, respectively. GC contents of *H. longispina aleutica*, *H. reniossa*, and *H. sanguinolenta* rRNAs were 37.4%, 37.7%, and 38.0%, respectively (Table 2). The 12S rRNA gene was positioned between two tRNA genes, phenylalanine, and glutamic acid, while the 16S rRNA gene resided between NADH2 and the D-loop region of *H. longispina aleutica*, *H. reniossa*, and *H. sanguinolenta* (Figure 1, Figure 2 and Figure 3, Table 2).

A total of 22 tRNA genes were identified within the three *Henricia* mitogenomes, 11 of which were encoded on the positive strand. Lengths of these tRNA genes ranged from 67 bp to 72 bp, with tRNA-Arg being the shortest and tRNA-Cys being the longest in *H. longispina aleutica*. In *H. reniossa*, tRNA genes ranged from 67 bp to 73 bp, with tRNA-Arg being the shortest and tRNA-Cys being the longest. Similarly, in *H. sanguinolenta*, tRNA-Lys was the shortest at 57 bp, while tRNA-Ile was the longest at 73 bp (Table 2). *H. longispina aleutica*, *H. reniossa*, and *H. sanguinolenta* tRNAs had GC contents of 35.1%, 36.2%, and 35.5%, respectively.

### 2.5. Phylogenetic Relationships and Gene Arrangement

Within class Asteroidea, a clade consisting of 42 species from six orders and 23 families was observed, confirming the systematic classification (Figure 4). The phylogenetic tree displayed a distinct separation of 42 asteroid mitogenomes from the outgroup. The three *Henricia* species in this study (*H. longispina aleutica*, *H. reniossa*, and *H. sanguinolenta*) belonged to the clade of family Echinasteridae, forming a monophyletic group alongside other *Henrica* species (*H. leviuscula* and *H. pachyderma*) and *Echinaster brasiliensis* from the same family. Within this clade, *H. reniossa* was a sister taxon of *H. leviuscula*, with a 100% bootstrap support value. Similarly, *H. longispina* aleutica was a sister taxon of *H. sanguinolenta*, with a 100% bootstrap support value as well (Figure 4).

## 3. Materials and Methods

### 3.1. Sample Collection

*Henricia* specimens were collected from several locations in the East Sea, Korea. They were collected by trimix and scuba diving (Table 4). All specimens were preserved in an ethyl alcohol solution (>95%) immediately after collection. These preserved specimens were stored at the Marine Echinoderms Resources Bank of Korea (MERBK) at Sahmyook University and assigned voucher numbers (Table 4).

### 3.2. Species Identification Based on the Morphological Characteristics

The genus *Henricia* Gray, 1840, in Korea, was primarily classified into two groups: the imbricated group and the reticulated group [14]. Among the specimens examined in this study, *H. sanguinolenta* (O.F. Müller, 1776) and *H. longispina aleutica* Fisher, 1911, were categorized under the reticulated group, while *H. reniossa asiatica* Djakonov, 1958, was assigned to the imbricated group. Subsequently, detailed morphological analyses were conducted on the studied specimens, comparing them with closely related species. The slender-armed *H. longispina aleutica* was compared to *H. pacifica* Hayashi, 1940, to elucidate morphological similarities and differences. Additionally, variations in the characteristics of *H. reniossa asiatica* were carefully examined in comparison to *H. reniossa reniossa* Hayashi, 1940. For further details on the examination of the studied specimens, refer to [14,20,21].

### 3.3. DNA Extraction and Mitochondrial DNA Amplification

Genomic DNAs were extracted from gonadal tissues using a DNeasy Blood & Tissue Kit (Qiagen, Hilden, Germany) according to the manufacturer’s protocol. Mitochondrial DNA amplification and analysis followed the method described by Lee and Shin [19]. Mitochondrial DNA amplification was done using the REPLI-g Mitochondrial DNA Kit (Qiagen, Hilden, Germany) according to the manufacturer’s protocol. The quality and concentration of amplified DNAs were assessed using a Nanodrop One-C (Thermo Fisher Scientific, Waltham, MA, USA). Next-generation sequencing (NGS) analysis was performed using genome analysis units at the National Instrumentation Center for Environmental Management at Seoul National University in Korea. A genomic library was constructed from the genomic DNA using a Kapa Hyper Prep Kit (Kapa Biosystems, Woburn, MA, USA), using paired-end reading, which was followed by NGS on the Illumina Hi-Seq 2500 platform (San Diego, CA, USA). The contigs of the mitogenome were assembled using the de novo assembly method on Geneious Prime 2023.1.1 (Biomatters Ltd., Auckland, New Zealand).

### 3.4. Mitogenome Annotation and Sequence Analysis

Complete mitogenomes of *H. longispina aleutica*, *H. reniossa*, and *H. sanguinolenta* were annotated by referencing three mitogenomes of closely related taxa from the family Echinasteridae available in the National Center for Biotechnology Information (NCBI) database (Table 5). Contigs of the mitogenome were assembled using the de novo assembly method with Geneious v.11.1.5 (Biomatters Ltd., Auckland, New Zealand). Subsequently, the complete mitogenome was annotated using the same software. tRNA genes were identified using the ARWEN website (http://130.235.244.92/ARWEN/ (accessed on 1 February 2024)) [20] and tRNAscan-SE online [21] using the “Invertebrate Mito” search mode. Geneious v.11.1.5 was also used to calculate the DNA base composition and codon preference of *H. longispina aleutica*, *H. reniossa*, and *H. sanguinolenta* mitogenomes. DNA base preference was determined using the following formulas: GC-skew = (G − C)/(G + C) and AT-skew = (A − T)/(A + T).

### 3.5. Phylogenetic Analysis

Phylogenetic analysis of the 13 PCGs in this study utilized the maximum likelihood (ML) method through RAxML 8.2 [43]. The dataset used for analysis consisted of 41 complete mitogenomes obtained from NCBI. This dataset included 42 species of class Asteroidea, including *H. longispina aleutica*, *H. reniossa*, and *H. sanguinolenta*, and two species of class Ophiuroidea, which served as the outgroup (Table 2). Alignment of mitogenome sequences was done using MAFFT [44]. The best-fit substitution model for the nucleotide dataset comprising 13 PCGs was determined using jModelTest 2.1.1 [45,46]. The selected substitution model for this dataset was GTR + I + G. For ML analysis, bootstrap resampling with 1000 iterations was conducted using the rapid option.

## 4. Discussion

Phylogenetic analysis is a valuable tool for identifying biological characteristics and studying species relationships and evolutionary history [47]. In previous investigations, morphological analyses were performed to identify species and determine evolutionary relationships. Subsequently, morphological analysis has been found to be insufficient for identifying species because morphological properties change faster than molecular analysis in response to changing geographical factors, environment, and climate. In addition, the damage may happen to the specimen’s body [48]. Therefore, mitochondrial DNA analysis, widely used in phylogenetic studies [49], was performed in this study to identify species. Due to its faster evolutionary rate than nuclear genetic markers, mtDNA is highly polymorphic. Thus, it can be used effectively as a DNA marker for breed identification [50].

*Henricia* species are spread over a wide range of geographic locations, meaning that different populations may encounter different environmental conditions. This can result in variations in physical characteristics, making it difficult to identify species within the genus. A combination of morphological and molecular phylogeny studies can be used for species identification to overcome this challenge.

Chichvarkhin [51] discovered seven different species of *Henricia* in Vostok Bay and two species from an adjacent area. These species were identified using morphological characteristics and DNA barcoding, which utilized partial mitochondrial gene (16S rRNA) sequences. Phylogenetic relationships of 17 species (*H. alexeyi*, *H. compacta*, *H. densispina*, *H. djakonovi*, *H. granulifera*, *H. hayashii*, *H. lineata*, *H. nipponica*, *H. obesa*, *H. oculate*, *H. ohshimai*, *H. olga*, *H. pacifica*, *H. sanguinolenta*, *H*. *sp*, *H. tumida*, and *H. uluudax*) and 39 specimens were determined using the neighbor-joining method (NJ). Results indicated that *H. sanguinolenta* formed a monophyletic clade and a large clade with *H. hayashii*, *H. obesa*, and *H. compacta*, with a 63% bootstrap support value. Additionally, *H. djakonovi* demonstrated a close relationship with *H. tumida*, with a 67% bootstrap support value. However, Ubagan [52] studied the molecular phylogeny and morphological characteristics of the species *Henricia djakonovi* using ten species of the genus *Henricia* (*H. epiphysialis*, *H. leviuscula*, *H. nipponica*, *H. oculata*, *H. perforate*, *H*. *regularis*, *H. reniossa*, *H. reticulata*, *H. sanguinolenta*, and *H. djakonovi).* Phylogenetic relationships were identified using ML based on mitochondrial DNA gene cytochrome c oxidase subunit I (COI). Results showed that *H. djakonovi* formed a monophyletic clade with a 78% bootstrap support value, while *H. sanguinolenta* showed a sister relationship with *H. regularis*, with low bootstrap support value. *H. reniossa* showed a sister relationship with *H. nipponica*, with a low bootstrap support value. *H. leviuscula* had a monophyly clade with a 100% bootstrap support value.

Scientists often use a partial gene of the mitochondrial genome, such as 16S rRNA and COI, to establish evolutionary relationships between species. However, results from these methods can differ and relationships between species can be unstable. Despite this, mitochondrial DNA is crucial for understanding evolutionary biology and tracing relationships between populations [1]. In this study, we successfully obtained the first complete mitochondrial genomes for *H. longispina aleutica*, *H. reniossa*, and *H. sanguinolenta* with GenBank accession numbers of PP384217, PP384218, and PP384219, respectively.

Lee and Shin [31] examined the complete mitochondrial genome of *H. leviuscula.* They reported phylogenetic relationships using ML based on 13 PCGs from 11 species of class Asteroidea (*Acanthaster brevispinus*, *A. planci*, *Aphelasterias japonica*, *Asterias amurensis*, *Echinaster (othilia) brasiliensis*, *Aquilonastra bather*, *Distolasterias nipon*, *H. leviuscula*, *Luidia quniaria*, *Patiria pectinifera*, and *Astropecten polyacanthus*). Results indicated that *H. leviuscula* had a close relationship with *E. brasiliensis*, with a 100% bootstrap support value. Similarly, the phylogenetic relationship of *H. pachyderma* has been analyzed using the ML approach based on 13 PCGs of 32 complete mitochondrial genomes of echinoderms. Of them, 16 contained asteroids. Results showed that *H. pachyderma* was more closely related to *H. leviuscula* than *E. brasiliensis*, with a strong bootstrap support value of 100%. It formed a large clade with *E. brasiliensis* with a 92% bootstrap support value [32].

As part of our investigation, we analyzed the phylogenetic relationships of 42 species belonging to the class Asteroidea, which included six species from the family Echinasteridae (*E. brasiliensis*, *H. leviuscula*, *H. longispina aleutica*, *H. pachyderma*, *H. reniossa*, and *H. sanguinolenta*). We performed ML analysis based on 13 PCGs. Results showed that *H. longispina aleutica* was closely related to *H. sanguinolenta* with a 100% bootstrap support value. Similarly, *H. reniossa* was closely related to *H. leviuscula* with a 100% bootstrap support value. The family Echinasteridae was identified as a monophyletic clade separate from the remaining families within the class Asteroid (Figure 4). Furthermore, the molecular study showed that *Henricia* species also formed a monophyletic clade within the family Echinasteridae. While morphologically similar, the seastar genus *Henricia* Gray, 1840, was more complex than other asteroid groups due to its highly variable characteristics, ability to interbreed, and production of intermediates [53,54]. The main diagnostic characteristics of *Henricia* can be determined by examining its abactinal and actinal morphological features. This includes observing the shape and number of abactinal and actinal spines, the structure of abactinal and actinal skeletons, and the number of ambulacral spines present. The abactinal skeleton of some *Henricia* species is typically arranged in a closely interlocking pattern, whereas others have a more open, reticulated structure [55]. Therefore, morphological features can vary within *Henricia* species. *H. longispina aleutica* Fisher, 1911 has a denuded abactinal spine. Its abactinal skeleton consists of lobed and elongated shapes that come together to form a rhomboid coarse meshwork of different sizes [20]. *H. reniossa* Hayashi, 1940, has a closely meshed dorsal skeleton with five or six long and stout inner spines. Its abactinal plates are arranged in several longitudinal series exclusively at the proximal abactinolateral section of the arms. *H. sanguinolenta* Fisher, 1911, has a weakly reticulated structure with narrow meshes in its abactinal skeleton. Its abactinal plate is small, with one to five abactinal papulae [20].

Findings from previous studies and mitogenomes uncovered in this research augment existing genomic resources for use in further evolutionary research on the Asteroidea class and beyond. These findings will be instrumental in conservation genetics.

## 5. Conclusions

In this study, we successfully determined complete mitogenomes and investigated the phylogenetic relationships of three species in the genus *Henricia*: *H. longispina aleutica*, *H. reniossa*, and *H. sanguinolenta*. Mitogenomes of these species comprised 13 protein-coding genes (PCGs), two rRNA genes, 22 tRNA genes, and a non-coding region. The total lengths of these complete mitogenomes were 16,217, 16,223, and 16,194 bp, respectively, consistent with reported lengths of other published echinoderm species. This study provides valuable insight into the phylogenetic and evolutionary relationships of genus *Henricia* and other echinoderm species. Additionally, the complete mitochondrial genomes of this study could be used to identify *H. longispina aleutica*, *H. reniossa*, and *H. sanguinolenta* more easily and provide further support for morphological identification under limited information by offering insight gained from mtDNA analysis.

## Figures and Tables

**Figure 1 ijms-25-05575-f001:**
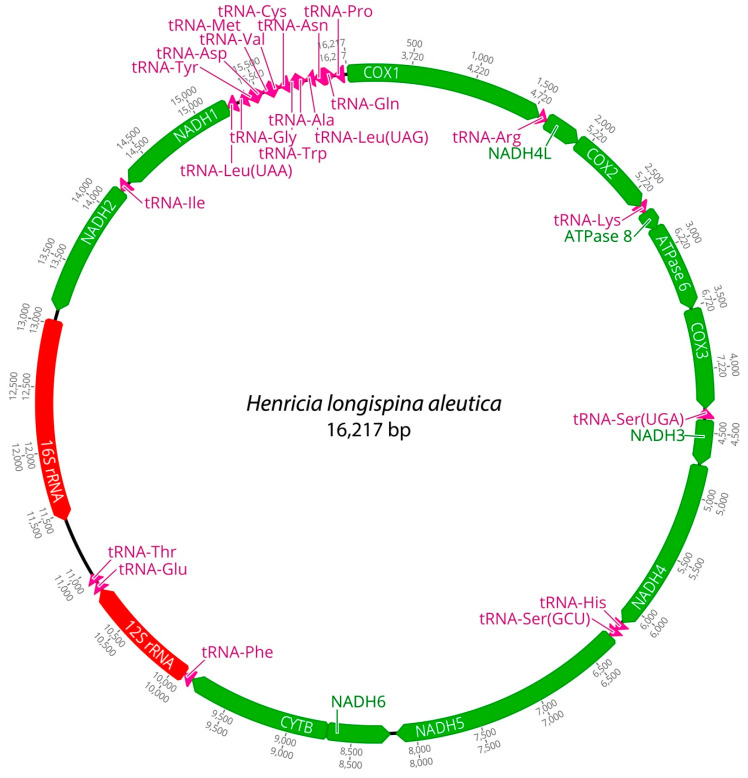
Complete mitochondrial genome of *Henricia longispina aleutica* in this study.

**Figure 2 ijms-25-05575-f002:**
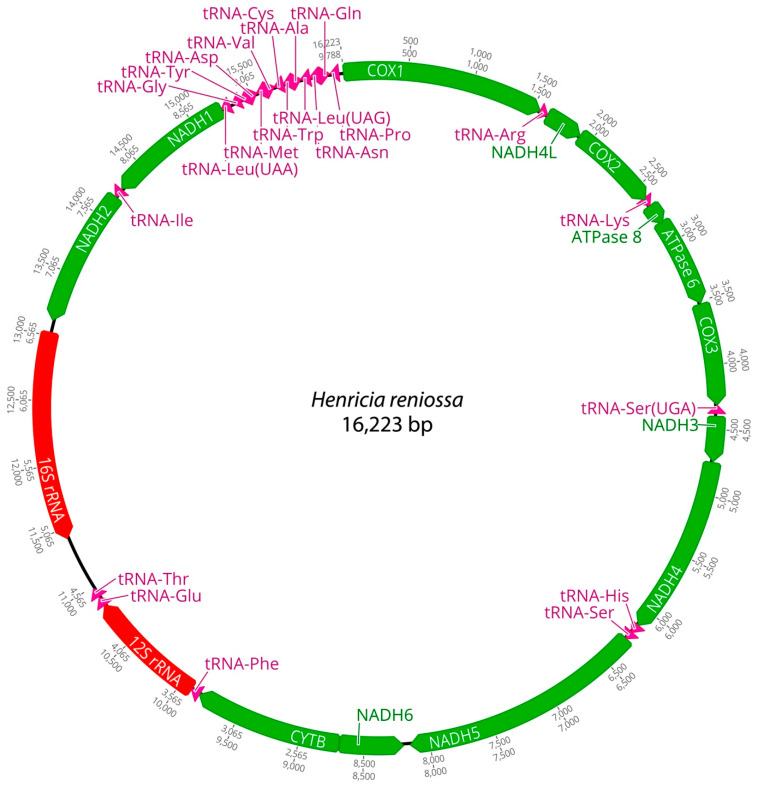
Complete mitochondrial genome of *Henricia reniossa* in this study.

**Figure 3 ijms-25-05575-f003:**
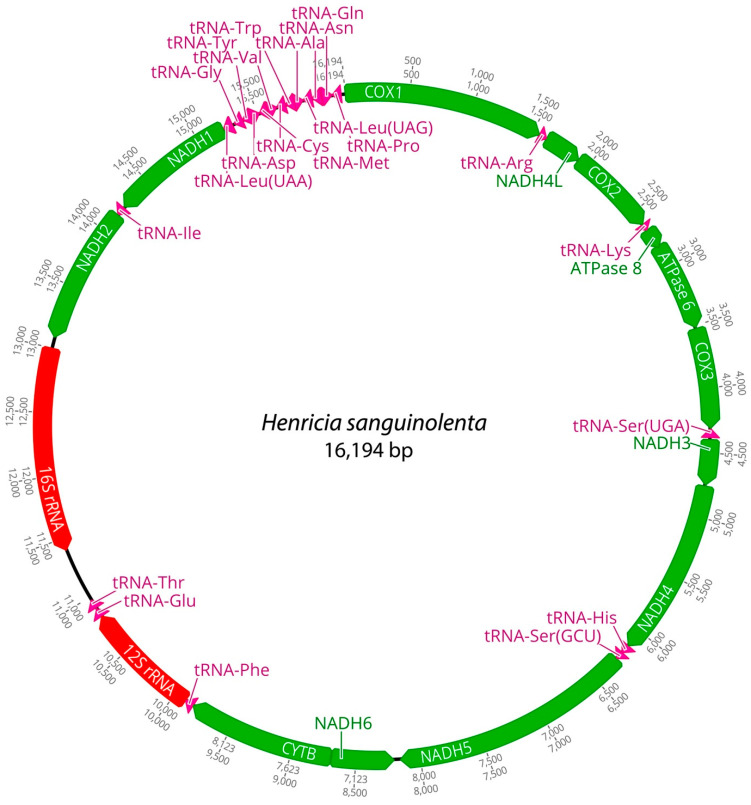
Complete mitochondrial DNA of *Henricia sanguinolenta* in this study.

**Figure 4 ijms-25-05575-f004:**
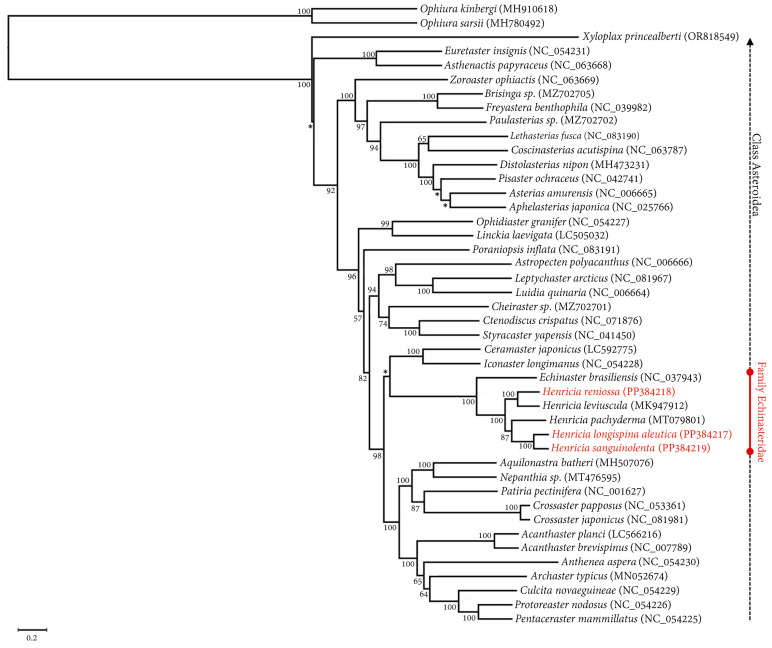
The maximum likelihood tree of the 3 *Henricia* species of this study, 42 asteroid species, and two ophiuroids as an outgroup based on nucleotide sequences of 13 PCGs. The asterisk (*) denotes a bootstrap support value under 50%.

**Table 1 ijms-25-05575-t001:** Nucleotide compositions of complete mitochondrial genomes of the three *Henricia* species in this study.

Species	Region	Size (bp)	A%	G%	C%	T%	AT%	GC%	ATskew	GCskew
*Henricia longispina aleutica*	Mitogenome	16,217	36.2	13.5	24.1	26.2	62.4	37.5	0.15	−0.28
	PCGs	11,473	36.3	13.1	24.9	25.6	61.9	38.0	0.17	−0.30
	tRNAs	1537	35.7	14.1	21.1	29.1	64.8	35.1	0.10	−0.19
	rRNAs	2516	37.3	13.9	23.5	25.3	62.5	37.4	0.19	−0.25
*Henricia reniossa*	Mitogenome	16,223	35.4	13.9	25.0	25.7	61.1	38.8	0.15	−0.28
	PCGs	11,476	35.3	14.0	25.8	25.0	60.2	39.7	0.17	−0.29
	tRNAs	1541	34.9	14.7	21.6	28.8	63.7	36.2	0.09	−0.19
	rRNAs	2535	37.1	13.8	24.0	25.1	51.4	37.7	0.19	−0.26
*Henricia sanguinolenta*	Mitogenome	16,194	35.6	13.9	24.6	25.8	61.4	38.5	0.15	−0.27
	PCGs	11,473	35.5	13.9	25.5	25.1	60.6	39.3	0.17	−0.29
	tRNAs	1531	35.5	14.2	21.3	29.0	64.4	35.5	0.10	−0.19
	rRNAs	2516	37.1	14.1	24.0	24.8	61.9	38.0	0.19	−0.25

**Table 2 ijms-25-05575-t002:** Mitochondrial genes and associated features of the three *Henricia* species in this study (intergenic space (IGS) is described as intergenic (+) or overlapping nucleotides).

Species	Gene	Type	Strand	Amino Acids	Start	Stop	Length (bp)	Start	Stop	Anti-Codon
*Henricia longispina aleutica*	COX1	PCG	H	518	1	1554	1554	ATG	TGA	
	tRNA-Arg	tRNA	H		1555	1621	67			CCU
	NAD4L	PCG	H	98	1622	1915	294	ATC	TAA	
	COX2	PCG	H	234	1916	2617	702	ATG	TAA	
	tRNA-Lys	tRNA	H		2604	2675	72			GAA
	ATP8	PCG	H	55	2676	2840	165	ATG	TAA	
	ATP6	PCG	H	231	2825	3517	693	ATG	TAA	
	COX3	PCG	H	261	3524	4306	783	ATG	TAA	
	tRNA-Ser (UGA)	tRNA	L		4312	4382	71			GAA
	NADH3	PCG	H	117	4398	4748	351	ATT	TAA	
	NADH4	PCG	H	461	4753	6135	1383	ATG	TGA	
	tRNA-His	tRNA	H		6140	6206	67			GUC
	tRNA-Ser (GCU)	tRNA	H		6207	6275	69			UCG
	NADH5	PCG	H	626	6276	8153	1878	ATG	TAA	
	NADH6	PCG	L	163	8200	8688	489	ATG	TAA	
	CYTB	PCG	H	379	8697	9834	1138	ATG	TAG	
	tRNA-Phe	tRNA	H		9835	9905	71			GCA
	12S rRNA	rRNA	H		9906	10,836	931			
	tRNA-Glu	tRNA	H		10,837	10,905	69			AUC
	tRNA-Thr	tRNA	H		10,909	10,979	71			AUU
	16S rRNA	rRNA	L		11,444	13,028	1585			
	NADH2	PCG	L	355	13,116	14,180	1065	ATG	TGA	
	tRNA-Ile	tRNA	L		14,181	14,251	71			AAA
	NADH1	PCG	L	326	14,265	15,242	978	ATG	TAA	
	tRNA-Leu (UAA)	tRNA	L		15,243	15,314	72			AAC
	tRNA-Gly	tRNA	L		15,336	15,403	68			UAU
	tRNA-Tyr	tRNA	L		15,404	15,472	69			UAA
	tRNA-Asp	tRNA	H		15,473	15,539	67			UCU
	tRNA-Met	tRNA	L		15,540	15,608	69			AAA
	tRNA-Val	tRNA	H		15,614	15,618	68			CGA
	tRNA-Cys	tRNA	L		15,680	15,751	72			GAA
	tRNA-Trp	tRNA	L		15,752	15,821	70			CCA
	tRNA-Ala	tRNA	H		15,822	15,889	68			ACA
	tRNA-Leu (UAG)	tRNA	L		15,890	15,960	71			AUU
	tRNA-Asn	tRNA	L		15,961	16,031	72			CUG
	tRNA-Gln	tRNA	H		16,043	16,113	71			UAA
	tRNA-Pro	tRNA	L		16,117	16,188	72			UCA
	D-loop	-	H		10,982	11,442	460			
*Henricia reniossa*	COX1	PCG	H	518	1	1554	1554	ATG	TGA	
	tRNA-Arg	tRNA	H		1555	1621	67			CCU
	NAD4L	PCG	H	98	1622	1915	294	ATC	TAA	
	COX2	PCG	H	234	1916	2617	702	ATG	TAA	
	tRNA-Lys	tRNA	H		2604	2674	71			GGA
	ATP8	PCG	H	55	2676	2840	165	ATG	TAA	
	ATP6	PCG	H	231	2825	3517	693	ATG	TAA	
	COX3	PCG	H	261	3525	4307	783	ATG	TAA	
	tRNA-Ser (UGA)	tRNA	L		4308	4377	70			GAG
	NADH3	PCG	H	117	4394	4744	351	ATT	TAA	
	NADH4	PCG	H	461	4749	6131	1383	ATG	TAA	
	tRNA-His	tRNA	H		6136	6202	67			GUC
	tRNA-Ser (GCU)	tRNA	H		6203	6271	69			UCG
	NADH5	PCG	H	627	6272	8152	1881	ATG	TAA	
	NADH6	PCG	L	163	8204	8692	489	ATG	TAA	
	CYTB	PCG	H	379	8701	9838	1138	ATG	TGA	
	tRNA-Phe	tRNA	H		9839	9910	72			GCA
	12S rRNA	rRNA	H		9911	10,838	928			
	tRNA-Glu	tRNA	H		10,839	10,910	72			AUU
	tRNA-Thr	tRNA	H		10,918	10,987	70			CUU
	16S rRNA	rRNA	L		11,420	13,026	1607			
	NADH2	PCG	L	355	13,121	14,185	1065	GTG	TAA	
	tRNA-Ile	tRNA	L		14,186	14,256	71			AAA
	NADH1	PCG	L	326	14,269	15,246	978	GTG	TAA	
	tRNA-Leu (UAA)	tRNA	L		15,247	15,318	72			AAC
	tRNA-Gly	tRNA	L		15,339	15,406	68			UAU
	tRNA-Tyr	tRNA	L		15,407	15,475	69			UAA
	tRNA-Asp	tRNA	H		15,476	15,544	69			CUA
	tRNA-Met	tRNA	L		15,545	15,612	68			AAC
	tRNA-Val	tRNA	H		15,618	15,686	69			CGA
	tRNA-Cys	tRNA	L		15,685	15,757	73			GAA
	tRNA-Trp	tRNA	L		15,758	15,827	70			CAG
	tRNA-Ala	tRNA	H		15,828	15,896	69			ACA
	tRNA-Leu (UAG)	tRNA	L		15,897	15,967	71			GUU
	tRNA-Asn	tRNA	L		15,970	16,041	72			CUG
	tRNA-Gln	tRNA	H		16,050	16,120	71			UAA
	tRNA-Pro	tRNA	L		16,123	16,193	71			UCA
	D-loop	-	H		10,989	11,417	428			
*Henricia sanguinolenta*	COX1	PCG	H	518	1	1554	1554	ATG	TGA	
	tRNA-Arg	tRNA	H		1555	1621	67			CCU
	NAD4L	PCG	H	98	1622	1915	294	ATC	TAA	
	COX2	PCG	H	234	1916	2617	702	ATG	TAA	
	tRNA-Lys	tRNA	H		2618	2674	57			GGA
	ATP8	PCG	H	55	2676	2840	165	ATG	TAA	
	ATP6	PCG	H	231	2825	3517	693	ATG	TAA	
	COX3	PCG	H	261	3524	4306	783	ATG	TAA	
	tRNA-Ser (UGA)	tRNA	L		4312	4382	71			AAG
	NADH3	PCG	H	117	4398	4748	351	ATT	TAA	
	NADH4	PCG	H	461	4753	6135	1383	ATG	TGA	
	tRNA-His	tRNA	H		6140	6206	67			GUC
	tRNA-Ser (GCU)	tRNA	H		6207	6275	69			UCG
	NADH5	PCG	H	626	6276	8153	1878	ATG	TAA	
	NADH6	PCG	L	163	8203	8691	489	ATG	TAA	
	CYTB	PCG	H	379	8700	9837	1138	ATG	TAG	
	tRNA-Phe	tRNA	H		9838	9909	72			GCA
	12S rRNA	rRNA	H		9910	10,831	922			
	tRNA-Glu	tRNA	H		10,832	10,903	72			AUC
	tRNA-Thr	tRNA	H		10,908	10,977	70			GAU
	16S rRNA	rRNA	L		11,407	13,000	1594			
	NADH2	PCG	L	355	13,087	14,151	1065	ATG	TAA	
	tRNA-Ile	tRNA	L		14,152	14,224	73			AAA
	NADH1	PCG	L	326	14,236	15,213	978	ATG	TAA	
	tRNA-Leu (UAA)	tRNA	L		15,214	15,285	72			AAC
	tRNA-Gly	tRNA	L		15,307	15,374	68			UAU
	tRNA-Tyr	tRNA	L		15,375	15,443	69			UAA
	tRNA-Asp	tRNA	H		15,444	15,512	69			CUA
	tRNA-Met	tRNA	L		15,512	15,580	69			AAA
	tRNA-Val	tRNA	H		15,586	15,654	69			AGA
	tRNA-Cys	tRNA	L		15,653	15,724	72			GAA
	tRNA-Trp	tRNA	L		15,726	15,794	69			CAG
	tRNA-Ala	tRNA	H		15,795	15,864	70			ACA
	tRNA-Leu (UAG)	tRNA	L		15,865	15,935	71			GUU
	tRNA-Asn	tRNA	L		15,938	16,009	72			CUG
	tRNA-Gln	tRNA	H		16,020	16,090	71			UAA
	tRNA-Pro	tRNA	L		16,093	16,164	72			AUC
	D-loop	-	H		10,980	11,404	424			

Abbreviations: H, heavy chain; L, light chain; PCG, protein-coding gene; rRNA, ribosomal RNA gene; tRNA, transfer RNA gene.

**Table 3 ijms-25-05575-t003:** Results from the relative synonymous codon usage (RSCU) analysis for PCGs of the mitochondrial genomes of the three *Henricia* species in this study.

*Henricia longispina aleutica*								*Henricia reniossa*								*Henricia sanguinolenta*							
AA	Codon	Count	RSCU	AA	Codon	Count	RSCU	AA	Codon	Count	RSCU	AA	Codon	Count	RSCU	AA	Codon	Count	RSCU	AA	Codon	Count	RSCU
Ala	GCA	60	1.36	Pro	CCA	105	1	Ala	GCA	36	1.12	Pro	CCA	104	1.16	Ala	GCA	53	1.48	Pro	CCA	86	0.54
	GCC	73	1.68		CCC	153	1.44		GCC	55	1.72		CCC	111	1.24		GCC	59	1.64		CCC	102	1.32
	GCG	9	0.2		CCG	43	0.4		GCG	6	0.16		CCG	31	0.4		GCG	6	0.16		CCG	21	0.24
	GCU	32	0.72		CCU	115	1.08		GCU	29	0.92		CCU	102	1.16		GCU	24	0.64		CCU	99	1.28
Cys	UGC	48	1.14	Gln	CAA	182	1.44	Cys	UGC	22	1.06	Gln	CAA	122	1.54	Cys	UGC	32	1.16	Gln	CAA	108	1.38
	UGU	35	0.84		CAG	68	0.54		UGU	19	0.92		CAG	36	0.44		UGU	23	0.82		CAG	47	0.6
Asp	GAC	57	0.92	Arg	CGA	39	0.56	Asp	GAC	48	1.04	Arg	CGA	38	1.12	Asp	GAC	49	1.02	Arg	CGA	35	0.96
	GAU	67	1.08		CGC	23	0.32		GAU	36	0.86		CGC	32	0.6		GAU	47	0.98		CGC	14	0.36
Glu	GAA	82	1.4		CGG	26	0.36	Glu	GAA	54	1.24		CGG	21	0.4	Glu	GAA	65	1.38		CGG	12	0.32
	GAG	35	0.6		CGU	23	0.32		GAG	33	0.73		CGU	11	0.2		GAG	28	0.6		CGU	15	0.4
Phe	UUC	96	0.78	Ser	UCA	104	1.6	Phe	UUC	92	1.04	Ser	UCA	55	1.44	Phe	UUC	79	0.88	Ser	UCA	55	1.68
	UUU	150	1.22		UCC	111	1.76		UUU	83	0.94		UCC	64	1.68		UUU	97	1.1		UCC	64	1.84
Gly	GGA	68	1.56		UCG	24	0.32	Gly	GGA	59	1.76		UCG	10	0.24	Gly	GGA	75	2.44		UCG	10	0.32
	GGC	33	0.76		UCU	100	1.52		GGC	27	0.8		UCU	63	1.68		GGC	19	0.6		UCU	63	1.68
	GGG	31	0.72		AGC	84	1.28		GGG	23	0.68		AGC	64	1.68		GGG	10	3.2		AGC	51	1.2
	GGU	40	0.92		AGU	82	1.28		GGU	23	0.68		AGU	37	0.96		GGU	18	0.54		AGU	45	1.04
His	CAC	93	1.06		AGA	111	3.2	His	CAC	66	0.86		AGA	67	2.56	His	CAC	59	0.96		AGA	52	2.8
	CAU	82	0.94		AGG	52	1.52		CAU	86	1.12		AGG	40	1.52		CAU	62	1.02		AGG	18	0.96
Ile	AUA	160	1.32	Thr	ACA	139	1.32	Ile	AUA	93	1.11	Thr	ACA	93	1.4	Ile	AUA	128	1.35	Thr	ACA	123	1.48
	AUC	71	0.57		ACC	125	1.2		AUC	68	0.81		ACC	69	1.04		AUC	58	0.6		ACC	93	1.12
	AUU	127	1.05		ACG	37	0.36		AUU	91	1.08		ACG	19	0.28		AUU	98	1.02		ACG	23	0.28
Lys	AAA	330	1.52		ACU	112	1.08	Lys	AAA	229	1.5		ACU	82	1.24	Lys	AAA	260	1.62		ACU	86	1.04
	AAG	103	0.46	Val	GUA	52	2		AAG	75	0.4	Val	GUA	49	1.6		AAG	58	0.36	Val	GUA	43	1.88
Leu	CUA	108	1.2		GUC	37	1.45	Leu	CUA	30	0.48		GUC	21	0.68	Leu	CUA	94	1.5		GUC	21	0.92
	CUC	74	0.78		GUG	12	0.45		CUC	82	1.26		GUG	11	0.36		CUC	45	0.72		GUG	8	0.36
	CUG	50	0.54		GUU	26	1		CUG	30	0.48		GUU	40	1.32		CUG	30	0.48		GUU	18	0.8
	CUU	115	1.26	Trp	UGG	46	1		CUU	82	1.26	Trp	UGG	31	1		CUU	71	1.14	Trp	UGG	38	1
	UUA	128	1.38	Tyr	UAC	89	0.8		UUA	76	1.2	Tyr	UAC	50	0.78		UUA	90	1.44	Tyr	UAC	79	0.98
	UUG	67	0.72		UAU	130	1.18		UUG	27	0.42		UAU	76	1.2		UUG	35	0.54		UAU	82	1.02
Met	AUG	64	1	Stop	UAA	196	1.77	Met	AUG	51	1	Stop	UAA	125	1.68	Met	AUG	48	1	Stop	UAA	113	1.5
Asn	AAC	178	1.08		UAG	80	0.72	Asn	AAC	133	0.96		UAG	51	0.69	Asn	AAC	116	1		UAG	60	0.78
	AAU	149	0.9		UGA	57	0.51		AAU	140	1.02		UGA	46	0.6		AAU	113	0.98		UGA	52	0.69

Abbreviation: AA, amino acid.

**Table 4 ijms-25-05575-t004:** Information collected and voucher number of *Henricia* species in this study.

Species	Collection Method	Collection Depth (m)	Collection Site (GPS)	Collection Date	MERBK Voucher Number
*Henricia longispina aleutica*	Trimix SCUBA diving	42	Ulleung island, Korea (37°14′58.2″ N, 131°52′1.3″ E)	23 August 2023	MERBK-A0093
*Henricia reniossa*	Netting	51	Yangyang, Korea (37°58′42.7″ N, 128°48′42.9″ E)	1 September 2022	MERBK-A0018
*Henricia sanguinolenta*	SCUBA Diving	26	Ulleung island, Korea (37°32′32.5″ N, 130°55′13.7″ E)	21 May 2023	MERBK-A0066

Abbreviation: MERBK, Marine Echinoderm Resource Bank of Korea.

**Table 5 ijms-25-05575-t005:** Mitochondrial genomes of the phylum Echinodermata in this study, including three newly reported mitogenomes of *Henricia longispina aleutica*, *H. reniossa*, and *H. sanguinolenta*.

	Class	Order	Family	Species	GenBank Accession No.	References
1	Asteroidea	Brisingida	Brisingidae	*Brisinga* sp.	MZ702705	[22]
2			Freyellidae	*Freyastera benthophila*	NC_039982	[23]
3		Forcipulatida	Asteriidae	*Asterias amurensis*	NC_006665	[24]
4				*Aphelasterias japonica*	NC_025766	[25]
5				*Coscinasterias acutispina*	NC_063787	[26]
6				*Distolasterias nipon*	MH473231	[27]
7				*Lethasterias fusca*	OR466204	Unpublished
8				*Pisaster ochraceus*	NC_042741	Unpublished
9			Paulasteriidae	*Paulasterias* sp.	MZ702702	[22]
10			Zoroasteridae	*Zoroaster ophiactis*	NC_063669	[22]
11		Paxillosida	Astropectinidae	*Astropecten polyacanthus*	NC_006666	[24]
12				*Leptychaster arcticus*	NC_081967	Unpublished
13			Benthopectinidae	*Cheiraster* sp.	MZ702701	[22]
14			Ctenodiscidae	*Ctenodiscus crispatus*	NC_071876	Unpublished
15			Luidiidae	*Luidia quinaria*	NC_006664	[24]
16			Porcellanasteridae	*Styracaster yapensis*	NC_041450	[28]
17		Peripodida	Xyloplacidae	*Xyloplax princealberti*	OR818549	[29]
18		Spinulosida	Echinasteridae	*Echinaster brasiliensis*	NC_037943	[30]
19				*Henricia leviuscula*	MK947912	[31]
20				*Henricia longispina aleutica*	PP384217	This study
21				*Henricia pachyderma*	MT079801	[32]
22				*Henricia reniossa*	PP384218	This study
23				*Henricia sanguinolenta*	PP384219	This study
24		Valvatida	Acanthasteridae	*Acanthaster brevispinus*	NC_007789	[33]
25				*Acanthaster planci*	LC566216	[34]
26			Archasteridae	*Archaster typicus*	MN052674	[35]
27			Asterinidae	*Aquilonastra batheri*	MH507076	[27]
28				*Nepanthia* sp.	MT476595	[36]
29				*Patiria pectinifera*	NC_001627	[37]
30			Goniasteridae	*Ceramaster japonicus*	LC592775	[38]
31				*Iconaster longimanus*	NC_054228	[36]
32			Ophidiasteridae	*Linckia laevigata*	LC505032	[39]
33				*Ophidiaster granifer*	NC_054227	[36]
34			Oreasteridae	*Anthenea aspera*	NC_054230	[36]
35				*Culcita novaeguineae*	NC_054229	[36]
36				*Pentaceraster mammillatus*	NC_054225	[36]
37				*Protoreaster nodosus*	NC_054226	[36]
38			Poraniidae	*Poraniopsis inflata*	NC_083191	[40]
39			Solasteridae	*Crossaster japonicus*	NC_081981	Unpublished
40				*Crossaster papposus*	NC_053361	Unpublished
41		Velatida	Myxasteridae	*Asthenactis papyraceus*	NC_063668	[22]
42			Pterasteridae	*Euretaster insignis*	NC_054231	[36]
43	Ophiuroidea	Ophiurida	Ophiuridae	*Ophiura sarsii*	MH780492	[41]
44				*Ophiura kinbergi*	MH910618	[42]

## Data Availability

The genome sequence data that support the findings of this study are openly available in GenBank of NCBI at https://www.ncbi.nlm.nih.gov (accessed on 20 February 2024), under accession nos. PP384217, PP384218, and PP384219.

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
