# Peer review of "Complete Mitochondrial Genomes and Phylogenetic Analysis of Genus Henricia (Asteroidea: Spinulosida: Echinasteridae)"

_ijms, 2024, doi:10.3390/ijms25115575_

Round 1

Reviewer 1 Report

Comments and Suggestions for Authors

Reviewer's report 

Date: 15 April 2024

Journal: International Journal of Molecular Sciences

Manuscript ID: ijms-2978389

Title: "Complete mitochondrial genomes and phylogenetic analysis of genus Henricia (Asteroidea: Spinulosida: Echinasteridae)"

Authors: Maria Alboasud et al. 

The authors have sequenced and investigated in detail the structure, organization, and evolution of complete mitochondrial genomes in three species of genus Henricia (Asteroidea). Combined with mitochondrial genomes from GenBank, they reconstruct phylogenetic relationships within the family Echinasteridae. 

The manuscript contains some inaccuracies. However, the data are sound, adequately described and illustrated, and that may provide important cues to scientists interested in thereby support the usage of mitogenome sequencing in echinoderm genetics. Therefore, after amending the manuscript according to the following suggestion, it is suitable for publication in the International Journal of Molecular Sciences.

Line 17: change “morphological characteristics” to “genetic characteristics”. 

Line 62: change “we adopted a multi-gene approach” to “we used a mitogenomic approach”. This change is necessary because due to limited recombination mitochondrial genome can be considered as a single locus. 

Line 67: change “inter- and intra-specific phylogenetic relationships” to inter-specific phylogenetic relationships. This change is necessary because the authors did not study intra-specific phylogenetic relationships.

Lines 172, 175, 177-179 should not be in Italycs.

Lines 198-239: the authors use only bootstrap values to characterize the relationships between the species. However, it is not enough. It is necessary to calculate at least p-distances between species. 

Author Response

Dear Reviewer 1,

Thank you for your thorough review and valuable comments on our article. We have carefully proofread the manuscript according to your suggestions and have incorporated all your feedback. You can find specific responses to each of your comments in the attached Word file.

Best regards,
Co-corresponding author

Reviewer 2 Report

Comments and Suggestions for Authors

The manuscript provides the analysis of three complete mitochondrial genomes of starfishes from the genus Henricia. The genus Henricia is known as a problematic for correct identification and phylogenetic reconstruction. Using the morphological characters only often leads to a wrong identification. In other words, I can approve: Identification of Henricia species is a nightmare! In this respect, the help of molecular characters, such as mitochondrial genome or even single genes may significantly facilitate the problem. Thus the presented manuscript provides important new data that will be of high demand by a number of specialists. Nevertheless, some issues found in the manuscript do not allow me to recommend this manuscript for publication in its current form. Below you will find the list of my comments and questions.

Major comments.

The introduction is well written, but I do not understand why the authors began with the information about mitogenomes? This is a very basic information that looks like textbook facts. It looks like a text that is needed just to inflate the manuscript to fit the word count. I’d recommend removing or significantly decreasing this paragraph.
In the Material and Methods section, a lot of important information is lacking. First of all, how did the authors perform the species identification? The photos or other morphological features of the studied specimens, though not necessary, would significantly improve the manuscript. Second, there is no information about library preparation and sequencing. I have taken a look at the publication by Lee and Shin (2018), but there is already about 6 years passed and Illumina HiSeq 2500 instrument is no longer supported by a manufacturer (according to the information on the official website https://support.illumina.com/sequencing/sequencing_instruments/hiseq_2500.html). Please, provide the full information. Third I would encourage the authors to provide some information about data availability (accession numbers and so on). I cannot understand why the first mention of Genbank accession numbers is at page 5, line 222!? This is a result and it should be mentioned at the beginning of the corresponding section.
In the discussion section, you mention the publications by Chichvarkhin et al. (2019) and Ubagan et al. (2023). Here the authors provided phylogenetic reconstructions based on single genes (16s rRNA and COI respectively). In my opinion, it would be of great importance to provide new single gene analyses with the addition of your data. Your phylogenetic analysis, though based on all the 13 PCGs, contains only 5 Henricia species which is far not sufficient. So I encourage the authors to make such analyses since it would significantly improve the quality of the study as well as the interest of specialists wolrldwide. I understand your opinion on such single-gene analyses, but since we still have such a drastic lack of complete mitochondrial genomes, the single-gene phylogeny is still a kind of standard. Another reason to include single gene analyses is the possibility of making the discussion of Chichvarkhin et al. (2019) and Ubagan et al. (2023) more comprehensive and interesting. These results may (or may not) provide more serious arguments for your opinion presented in lines 216-220.

Minor comments

Lines 170-179. There was some kind of formatting error.
Line 206. Ref #33 to Ubagan should be #32 according to the reference list.
Another minor question is about the figures 1-3. What software did you use to obtain these mitogenome maps?

Author Response

Dear Reviewer 2,

Thank you for your thorough review and valuable comments on our article. We have carefully proofread the manuscript according to your suggestions and have incorporated all your feedback. You can find specific responses to each of your comments in the attached Word file.

Best regards,
Co-corresponding author

Reviewer 3 Report

Comments and Suggestions for Authors

The manuscript by Alboasud et al. describes three complete mitochondrial genomes of the sea stars belonging to genus Henricia. They also construct a phylogeny of these species, combining their data with 41 mitogenomes from GenBank. Class Asteroidea (sea stars) is a very diverse group with 1800 species, but only a few dozen species have their mitogenomes completely sequenced, so this work is commendable as it helps to classify biodiversity and in species identification. The study is presented in a quite typical way for mitogenome reports. The authors describe genes, provide genetic maps, and also characterize nucleotide composition, codon usage bias, as well as atypical start- and stop codons of the protein coding genes (PCGs). In the discission, the authors review previously published publications that reported Henricia species. The authors also discuss morphological characters used to distinguish species and show that morphology is insufficient for the phylogeny reconstruction and can be unreliable for species identification. Altogether, the manuscript is clearly written and easy to understand, figures and tables are informative, introduction and discussion are satisfactory.

There are two major points that need to be addressed:

1) Currently, Materials and Methods do not describe the methods that were used for mitogenome sequencing (e.g. Sanger, Illumina, Nanopore, etc) and how the sequences were processed. If sequences were done using Next-Generation Sequencing methods, some results should be presented, explaining coverage, sequence quality, etc.

2) The section ‘Relative synonymous codon usage’ on lines 142-148 has to be rewritten after some corrections in the analysis. The authors seem to confuse RSCU (relative synonymous codon usage) and Codon Usage (the observed frequencies of specific codons). The RSCU is a Codon Usage adjusted for the number of codons for specific amino acids (hence ‘relative’). In Table 5, columns designated as ‘RSCU’ show, in fact, the Codon Usage. In order to calculate RSCU, the values in columns ‘RSCU’ have to be multiplied by the number of codons for specific amino acid. For example, Ala has 4 codons, and so RSCUs for H. longispina would be GCA: 0.34*4=1.38; GCC: 0.42*4=1.68; GCG: 0.05*4=0.2; GCU: 0.18*4=0.72. The total of RSCU for Ala should be equal 4 (there may be small departures due to the rounding error). Another example: Leu has 6 codons, and by multiplying values in the column ‘RSCU’ by 6 we obtain the following correct RSCUs for H. longispina:  CUA: 1.20; CUC: 0.82; CUG: 0.55; CUU: 1.27; UUA: 1.42; UUG: 0.74, with a total of 6. RSCU for Met and Trp is always 1, because these amino acids are each encoded by 1 codon.

Some additional specific points:

Line 35: Some statements on lines 31-35 are not supported by references. For example, substitution or mutation rates are not discussed in ref [1]: Boore 1999. Please provide additional References. Furthermore, substitution rates and mutation rates mean essentially the same thing in a given context. Use ‘substitution rate’ and delete ‘mutation rate’.

Line 39: The following sentence is not clear: “Its genes are arranged in series, enabling a greater number of informative sites”

Line 46: Change “Starfish are also renowned for their wide range of…” with “Starfish are also renowned as a source of wide range of compounds with…”

Line 51: Replace "are due to" with "are sometimes affected by"

Line 67: This study does not investigate “intra-specific” relationships; it only studies interspecific relationships. All data are represented by a single specimen (or mitogenome) for each species.

~ Line 83: Information is missing about what methods were used for mitogenome sequencing (e.g. Sanger, Illumina, Nanopore, etc) and how the sequences were processed.

Lines 101-102. It is necessary to tell that for this analysis only PCGs were aligned (not whole mitogenomes) and used for phylogeny reconstruction.

Lines 107-122. In this paragraph use “GC-skew” instead of “anti-G bias”

Lines 142-148. The section ‘Relative synonymous codon usage’ has to be rewritten – see above in major comments.

Lines 146-148: The values shown here should not have % sign (i.e., they should be 1.00, 0.04, 0.03). Multiply those values by 100 to show them as percentage (100%, 4%, 3%).

Line 173: Do you mean ‘Figure 4’?

Line 177-179: It will sound better if you say that H. reniossa was a sister-taxon of H. leviuscula, and H. longispina aleutica was a sister-taxon of H. sanguinolenta.

Line 199: Correct parentheses to include ‘rRNA’. i.e. (16S rRNA)

Line 223: Change ‘Lee’ to ‘Lee and Shin’

Line 228: E. brasiliensis discussed here is not in among 11 species listed on lines 225-227. On the other hand, Astopecten brasiliensis is listed, but it is not in the phylogeny by Lee and Shin (2019).

Line 241: It must be Figure 4, not Figure 2.

Line 241-242: The sentence could be modified as follows: “Furthermore, the molecular study showed that Henricia species also formed a monophyletic clade within family Echinasteridae.”

Line 242: Use lowercase letters for ‘genus’

References:

Line 282: All words in journal names should start with capital letters, e.g. ‘Journal of Heredity’. Please correct here and in other references, e.g. Ref #16, 21, 23, and others.

Line 391: Change O’HARA to O’Hara 

I could not find References ##39 and 40 in the main text. Please check that they are either cited or removed from References.

Author Response

Dear Reviewer 3,

Thank you for your thorough review and valuable comments on our article. We have carefully proofread the manuscript according to your suggestions and have incorporated all your feedback. You can find specific responses to each of your comments in the attached Word file.

Best regards,
Co-corresponding author

Round 2

Reviewer 2 Report

Comments and Suggestions for Authors

The new version of the manuscript has a number of improvements in the text, as well as more adequate formatting. The authors took into account some of my comments, BUT for some reason they did not respond at all to a number of my MAJOR comments. I would not mind if the authors had a different position on the comments I made, but I would like to see reasoned responses to my comments. It appears that my main comments were either forgotten or deliberately ignored.

Below I repeat the list of my major comments, that were not addressed.

The introduction is well written, but I do not understand why the authors began with the information about mitogenomes? This is a very basic information that looks like textbook facts. It looks like a text that is needed just to inflate the manuscript to fit the word count. I’d recommend removing or significantly decreasing this paragraph.

In the Material and Methods section, some important information is lacking. How did the authors perform the species identification? The photos or other morphological features of the studied specimens, though not necessary, would significantly improve the manuscript.

In the discussion section, you mention the publications by Chichvarkhin et al. (2019) and Ubagan et al. (2023). Here the authors provided phylogenetic reconstructions based on single genes (16s rRNA and COI respectively). In my opinion, it would be of great importance to provide new single gene analyses with the addition of your data. Your phylogenetic analysis, though based on all the 13 PCGs, contains only 5 Henricia species which is far not sufficient. So I encourage the authors to make such analyses since it would significantly improve the quality of the study as well as the interest of specialists wolrldwide. I understand your opinion on such single-gene analyses, but since we still have such a drastic lack of complete mitochondrial genomes, the single-gene phylogeny is still a kind of standard. Another reason to include single gene analyses is the possibility of making the discussion of Chichvarkhin et al. (2019) and Ubagan et al. (2023) more comprehensive and interesting. These results may (or may not) provide more serious arguments for your opinion presented in lines 216-220.

Author Response

[Response for the 2nd comment from REVIEWER 2]

Dear REVIEWER 2,

Firstly, I apologize for our previous response falling short of addressing your comment adequately.

I want to assure you that it was not intentional.

Below this letter, you'll find our response to your second reviewing comment.

Thank you for your valuable feedback, which will contribute to enhancing our manuscript.

Co-corresponding Author,

Taekjun Lee

---------------------------------------------------------------------------------------------------------------------------------------

[Point 1] The introduction is well written, but I do not understand why the authors began with the information about mitogenomes? This is a very basic information that looks like textbook facts. It looks like a text that is needed just to inflate the manuscript to fit the word count. I’d recommend removing or significantly decreasing this paragraph.

[Answer] I think it is important to talk a bit about mitogenome so I decreased it according to your comment [Line 24-33].

[Point 2] In the Material and Methods section, some important information is lacking. How did the authors perform the species identification? The photos or other morphological features of the studied specimens, though not necessary, would significantly improve the manuscript.

[Answer] It has been revised according to your comment. Thus, we newly described “Species identification based on the morphological characteristics” section in the Materials and methods part [Line 163-174].

[Point 3] In the discussion section, you mention the publications by Chichvarkhin et al. (2019) and Ubagan et al. (2023). Here the authors provided phylogenetic reconstructions based on single genes (16s rRNA and COI respectively). In my opinion, it would be of great importance to provide new single gene analyses with the addition of your data. Your phylogenetic analysis, though based on all the 13 PCGs, contains only 5 Henricia species which is far not sufficient. So I encourage the authors to make such analyses since it would significantly improve the quality of the study as well as the interest of specialists worldwide. I understand your opinion on such single-gene analyses, but since we still have such a drastic lack of complete mitochondrial genomes, the single-gene phylogeny is still a kind of standard. Another reason to include single gene analyses is the possibility of making the discussion of Chichvarkhin et al. (2019) and Ubagan et al. (2023) more comprehensive and interesting. These results may (or may not) provide more serious arguments for your opinion presented in lines 216-220.

[Answer] Thank you for your comment. We acknowledge that the representation of phylogenetic relationships in Henricia was limited with only 5 mitogenomes. To overcome this limitation, we obtained partial sequences of the 16S rRNA and COI genes from numerous Henricia specimens collected over the past three years. The analysis of these sequences has recently been completed, and we plan to submit the findings to another journal by June. Our primary focus in this manuscript is to expand and establish the phylogenetic relationships within Henricia and asteroids based on mitogenomes, which aligns with the theme of this special issue in the journal. We kindly request your understanding and support on this matter.

Round 3

Reviewer 2 Report

Comments and Suggestions for Authors

The authors addressed all my comments. I am satisfied with the responses and I have no more further questions. I wish you good luck with the publication.

Author Response

Dear Reviewer,

Thank you for your positive feedback and for acknowledging that we have addressed all your comments satisfactorily. We are pleased to hear that you have no further questions regarding the manuscript. Your well wishes for the publication are greatly appreciated.

Sincerely,
Co-corresponding Author